# ONE-STAGE DEEP FEATURE INSTRUMENTAL VARIABLE REGRESSION

## ABSTRACT

Instrumental variable (IV) regression is a fundamental tool for causal inference in the presence of unmeasured confounders. Traditional approaches, such as two-stage least squares and its modern extensions, rely on a two-stage learning procedure. However, this two-stage paradigm introduces inherent issues, such as error propagation, computational overhead, and sensitivity to model mis-specification. To solve these issues, we propose a novel one-stage deep feature IV (1SDFIV) regression method to directly learn the causal structural function, using a loss based on the measure of martingale difference divergence (MDD). Since MDD-type loss directly leverages the exogeneity of IV, 1SDFIV is endowed with a key characteristic of first-stage learning, bypassing the need to learn an exogenous variable from IV as required in those two-stage learning methods. Moreover, we further propose a generalized 1SDFIV (G-1SDFIV) regression method, which could achieve improved prediction performance when the causal model is mis-specified. Experimental results on both low- and high-dimensional settings show that 1SDFIV and G-1SDFIV significantly outperform their two-stage competitors, offering more accurate predictions, reduced computational burden, and greater computational stability.

## 1 INTRODUCTION

Endogeneity caused by unmeasured confounders makes traditional regression methods fail to identify causal effects correctly. To solve this problem, a fundamental tool is to use instrumental variable (IV) regression, which aims to find an exogenous variable from a given valid IV. For the IV regression, its classical learning strategy is two-stage least squares (2SLS), which first regresses the treatment on the IV to obtain predicted value (treated as a learned exogenous variable), and then uses this exogenous variable to estimate the causal effect on the outcome in a second-stage regression. Although effective under linearity, 2SLS lacks the flexibility to model complex nonlinear relationships commonly observed in modern datasets.

To address this limitation, a number of nonparametric and semiparametric IV methods have been developed. Early work in econometrics includes sieve IV (Chen & Pouzo, 2012) and kernel IV (Singh et al., 2019). The method of sieve IV uses a growing dictionary of basis functions to approximate the structural function, while that of kernel IV leverages reproducing kernel Hilbert spaces for a flexible function approximation. Despite enjoying strong theoretical guarantees, the practical performance of both methods is often limited by the use of pre-specified feature maps, which may not adequately capture the underlying data structure without careful hand-engineering.

A more recent line of work casts IV regression as a conditional density estimation problem. DeepIV (Hartford et al., 2017) is a prominent example that uses deep neural networks to model the conditional distribution of the treatment given instruments and covariates in the first stage, followed by integration over this distribution in the second stage to predict the outcome. This approach allows highly flexible modeling of both treatment and outcome processes and has been successfully applied to real-world counterfactual prediction tasks, such as estimating the effect of ad placement on user click-through rates. However, DeepIV suffers from two practical drawbacks. First, estimating full conditional densities (especially in high-dimensional treatment spaces) is statistically challenging and prone to high variance. Second, the integration step in the second stage can be computationally expensive and unstable.

To overcome these drawbacks, Deep Feature IV (DFIV) (Xu et al., 2021) has emerged as a state-of-the-art method that combines the stability of 2SLS with the representational power of deep learning. Rather than estimating the full conditional distribution of the treatment, DFIV learns feature maps for both the instrument and the treatment using deep neural networks, and performs 2SLS regression directly in the learned feature space. This avoids the need for an explicit density estimation while still enabling a rich and nonlinear modeling of the structural relationship. As a result, DFIV inherits the simplicity and robustness of 2SLS while achieving superior empirical performance on a range of benchmark tasks.

Nevertheless, both DeepIV and DFIV fundamentally adhere to a two-stage framework. This framework critically depends on accurately modeling either the conditional expectation or the conditional distribution of treatment in the first stage, a task that typically requires neural networks with carefully tuned hyperparameters and substantial computational resources. Even with meticulous tuning, inadequate first-stage fits can propagate errors to the second stage, compromising the validity of causal effect estimates. Moreover, neural architectures and hyperparameter choices suitable for one dataset or application may not generalize well to others, raising concerns about the robustness, universality, and computational efficiency of both methods across diverse empirical contexts.

In this paper, we introduce a new one-stage deep feature IV (1SDFIV) regression method to learn causal effects. The key idea of 1SDFIV is to directly estimate the causal model using a loss based on the measure of martingale difference divergence (MDD) (Shao & Zhang, 2014; Park et al., 2015). The MDD-type loss directly makes use of the exogeneity of IV, so it bypasses the need to learn an exogenous variable from IV as done in those two-stage learning methods. Owing to the nature of one-stage learning, 1SDFIV avoids the error propagation commonly existing in two-stage approaches such as DeepIV and DFIV, and it further reduces computational burdens with a weak reliance on neural network and hyperparameter tuning. In addition, under certain conditions, we offer a theoretical guarantee to 1SDFIV for identifying the average causal effect. Moreover, we propose a generalized 1SDFIV (G-1SDFIV) regression method, which aims to enhance prediction power in the presence of causal model mis-specification. Finally, from benchmark experiments, we find that when the dimension of confounders is low, both 1SDFIV and G-1SDFIV can substantially outperform their competitors; and when the dimension of confounders is high, the advantage of G-1SDFIV over other methods remains significant.

The remainder of this paper is organized as follows. Section 2 provides the causal model, reviews the existing two-stage learning methods, and introduces the MDD measure. Section 3 presents our one-stage learning procedures and provides a theoretical guarantee for identifying the average causal effect. Section 4 demonstrates the superior performance of our proposed methods on low- and high-dimensional benchmark datasets. Section 5 concludes the paper.

## 2 PRELIMINARIES

### 2.1 CAUSAL MODEL AND ITS TWO-STAGE LEARNING METHODS

Let $X \in \mathcal{X}$ denote the treatment variable and $Y \in \mathbb{R}$ denote the outcome variable. Consider the following causal model:

$$Y = g(X) + \varepsilon \text{ with } \mathbb{E}[\varepsilon] = 0 \text{ and } \mathbb{E}[\varepsilon|X] \neq 0, \tag{1}$$

where $g : \mathcal{X} \to \mathbb{R}$ is the structural function that characterizes the causal effect of $X$ on $Y$, and $\varepsilon$ is the error term containing unmeasured confounders $U$ that are also related to $X$. Due to the presence of $U$, $X$ is endogenous with $\mathbb{E}[\varepsilon|X] \neq 0$, so that the classical least squares estimator (LSE) cannot identify $g$. To overcome this deficiency, one can make use of a valid exogenous IV $Z \in \mathcal{Z}$ for identification purposes, where $Z$ satisfies a key assumption (Angrist et al., 1996; Xu et al., 2021):

**Assumption 1** *The conditional distribution $P(X|Z)$ is not constant in $Z$ and $\mathbb{E}[\varepsilon|Z] = 0$.*

Under Assumption 1, the structural function $g$ is identified by solving the following *integral equation*:

$$\mathbb{E}[Y|Z = z] = \mathbb{E}[g(X)|Z = z], \quad \forall z \in \mathcal{Z}. \tag{2}$$

This formulation motivates many popular estimation methods of $g$ in a two-stage framework, such as the classical linear 2SLS method (Theil, 1953), the nonparametric sieve IV (Chen & Pouzo, 2012)

and kernel IV (Singh et al., 2019) methods, and the network-based DeepIV (Hartford et al., 2017) and DFIV (Xu et al., 2021) methods. Specifically, this framework learns the conditional expectation $\mathbb{E}(X|Z)$ or the conditional distribution $P(X|Z)$ in the first stage, and then it regresses $Y$ on features derived from $X$ in the second stage, commonly using predictions $\hat{X}$ or densities $p(x|z)$.

Among all outlined methods in this framework, DFIV generally has better prediction performances than 2SLS, sieve IV, kernel IV, and DeepIV for complex tasks, due to its computational stability and representational power. For the DFIV, it considers a structural function $g(X) = u^\top \phi(X, \theta)$, where $\phi(X, \theta)$ is a neural network parametrized by $\theta$ and $u$ is a vector of unknown parameters. In such setting, the causal model (1) can be expressed as

$$Y = u^\top \phi(X, \theta) + \varepsilon. \tag{3}$$

Let $\{(X_i, Y_i, Z_i)\}_{i=1}^n$ be a sequence of data observations. By assuming a similar representation for the map from $Z$ to $X$, the two-stage learning procedure of DFIV is given as follows:

**Stage 1**: Consider a network $\psi(Z) = Vh(Z, \kappa)$, where $h(Z, \kappa)$ is a neural network parametrized by $\kappa$ and $V$ is a matrix of unknown parameters. Estimate $(V, \kappa)$ by $(\hat{V}, \hat{\kappa})$, which is the minimizer of the following objection function:

$$R_{1n}(V, \kappa) = \frac{1}{n} \sum_{i=1}^n \|X_i - Vh(Z_i, \kappa)\|^2 + \lambda_1 \|V\|^2, \tag{4}$$

where $\lambda_1 \geq 0$ is a hyperparameter.

**Stage 2**: Based on model (3) and the learned exogenous variable $\hat{X}_i = \hat{V}h(Z_i, \hat{\kappa})$ from Stage 1, estimate $(u, \theta)$ by $(\hat{u}, \hat{\theta})$, which is the minimizer of the following objection function:

$$R_{2n}(u, \theta) = \frac{1}{n} \sum_{i=1}^n \|Y_i - u^\top \phi(\hat{X}_i, \theta)\|^2 + \lambda_2 \|u\|^2, \tag{5}$$

where $\lambda_2 \geq 0$ is a hyperparameter.

After implementing the above procedure, we can estimate the causal effect of $X$ on $Y$ by $\hat{u}^\top \phi(X, \hat{\theta})$. Clearly, the two-stage learning procedure for DFIV inevitably faces the issue of error propagation, that is, the learning error generated by the first stage could largely affect the prediction of causal effect in the second stage. In addition, another less critical issue of DFIV is that it needs to specify a neural network in each stage, resulting in annoying computational burdens and the challenge of tuning parameters selection. Note that these two issues also exist in DeepIV.

## 2.2 MARTINGALE DIFFERENCE DIVERGENCE

To solve the aforementioned issues, we attempt to learn model (3) in a new one-stage framework, which relies on the measure of martingale difference divergence (MDD) in Shao & Zhang (2014) and Park et al. (2015). For two random variables $V \in \mathbb{R}$ and $U \in \mathbb{R}^q$ with $\mathbb{E}\|V\| < \infty$ and $\mathbb{E}\|U\| < \infty$, MDD is defined as

$$\text{MDD}(V|U)^2 = \frac{1}{c_q} \int_{\mathbb{R}^q} \frac{\|g_{V,U}(s) - g_V g_U(s)\|^2}{\|s\|^{1+q}} ds,$$

where $c_q = \pi^{(1+q)/2}/\Gamma((1+q)/2)$ is a constant, $g_{V,U}(s) = \mathbb{E}\left(Ve^{i\langle s,U\rangle}\right), g_V = \mathbb{E}(V)$, and $g_U(s) = \mathbb{E}\left(e^{i\langle s,U\rangle}\right)$ with $i$ being the imaginary unit. Although MDD is defined in terms of integration, it has an equivalent form in terms of expectation (Park et al., 2015):

$$\text{MDD}(V|U)^2 = -\mathbb{E}[(V - E(V))^\top (V' - E(V'))\|U - U'\|],$$

where $(V', U')$ is an i.i.d. copy of $(V, U)$. Using the above form, given a sequence of data observations $\{(V_i, U_i)\}_{i=1}^n$, we can easily obtain the empirical estimator of $\text{MDD}(V|U)^2$ defined by

$$\text{MDD}_n(V|U)^2 = -\frac{1}{n^2} \sum_{j=1}^n \sum_{k=1}^n (V_j - \bar{V})^\top (V_k - \bar{V})\|U_j - U_k\|, \tag{6}$$

where $\bar{V} = n^{-1} \sum_{i=1}^{n} V_i$.

As shown in Theorem 1 in Shao & Zhang (2014) and Proposition 3.1 in Park et al. (2015), we have $\mathrm{MDD}(V|U)^2 \geq 0$ and

$$\mathrm{MDD}(V|U)^2 = 0 \text{ if and only if } \mathbb{E}[V|U] = \mathbb{E}[V]. \tag{7}$$

See Song et al. (2025) for further details. The property in (7) indicates that MDD captures the mean dependence of $V$ and $U$, and it is key to deriving our one-stage learning procedure in the sequel.

## 3 ONE-STAGE DEEP FEATURE IV REGRESSION

### 3.1 LEARNING STRATEGIES

Note that when $\mathbb{E}[\varepsilon|Z] = 0$ under Assumption 1, we have $\mathrm{MDD}(\varepsilon|Z)^2 = 0$. Hence, under (3), we can directly learn $u$ and $\theta$ by minimizing the objective function:

$$L_n(u, \theta) = \mathrm{MDD}_n(Y - u^\top \phi(X, \theta)|Z)^2, \tag{8}$$

where $\mathrm{MDD}_n(\cdot|\cdot)^2$ is defined as in (6). Clearly, the above estimation method is one-stage without learning an exogenous variable $\hat{X}$. However, a direct joint optimization of $L_n(u, \theta)$ is very time-consuming. To relieve this issue, based on an initial estimator $\tilde{\theta}_*$ of $\theta$, we first estimate $u$ by $\tilde{u}$, which is the minimizer of the objective function $R_n(u, \tilde{\theta}_*)$, where

$$R_n(u, \theta) = \frac{1}{n} \sum_{i=1}^{n} \|Y_i - u^\top \phi(X_i, \theta)\|^2 + \lambda \|u\|^2, \tag{9}$$

with $\lambda \geq 0$ being a hyperparameter. Clearly, $R_n(u, \theta)$ in (9) is the same as $R_{2n}(u, \theta)$ in (5), except that $X_i$ replaces $\hat{X}_i$. We should mention that we use $R_n(u, \theta)$ to estimate $u$ for two reasons. First, $R_n(u, \theta)$ accommodates the mean squared error prediction criterion. Second, for a given $\theta$, $R_n(u, \theta)$ becomes the ridge regression objective function which has an analytic minimizer. With the estimator $\tilde{u}$, we then estimate $\theta$ by $\tilde{\theta}$, which is the minimizer of the objective function $L_n(\tilde{u}, \theta)$. As usual, $\theta$ is allowed to have a high dimension, so we apply the Adam algorithm to compute $\tilde{\theta}$; meanwhile, we update $\tilde{u}$ and $\tilde{\theta}$ iteratively until convergence. Due to the one-stage nature, this proposed learning method is termed as the one-stage deep feature IV (1SDFIV) regression. See Algorithm 1 for its details.

---

**Algorithm 1** 1SDFIV Regression

**Require:** Data $(X_i, Y_i, Z_i)$; Regularization parameter $\lambda$; Initial estimator $\tilde{\theta}_*$. Batch size $n_b$; Learning rate $\alpha$; Number of updates $T$
**Ensure:** Estimated structural function $g(x)$
1: $\tilde{\theta} \leftarrow \tilde{\theta}_*$
2: **repeat**
3:      Sample $n_b$ data $(X_i^{(b)}, Y_i^{(b)}, Z_i^{(b)})$.
4:      **for** $t = 1$ to $T$ **do**
5:          Return $\tilde{u}^{(n_b)}$, i.e., the minimizer of $R_n^{(n_b)}(u, \tilde{\theta})$ using $(X_i^{(b)}, Y_i^{(b)})$
6:          Update $\tilde{\theta} \leftarrow \tilde{\theta} - \alpha \nabla_\theta L_n^{(n_b)}(\tilde{u}^{(n_b)}, \tilde{\theta})$
7:      **end for**
8: **until** convergence
9: Compute $\tilde{u} := \tilde{u}^{(n)}$, the minimizer of $R_n(u, \tilde{\theta})$ using entire dataset
10: **return** $\tilde{g}(x) = \tilde{u}^\top \phi(x, \tilde{\theta})$

---

Needless to say, when the sample size $n$ is small, the learned causal effect $\tilde{u}^\top \phi(X_i, \tilde{\theta})$ may not have the capacity to adequately approximate the true causal effect, which can have a correct specification different from that of model (3). In this case, using the loss of $\mathrm{MDD}_n(\cdot|\cdot)^2$ along could lead to inferior predictions under the mean squared error criterion. To overcome this issue, we consider a

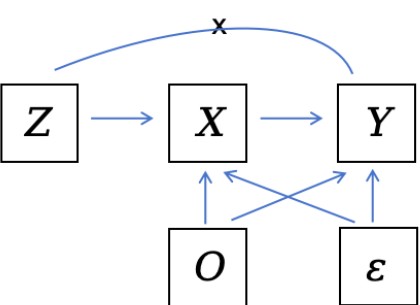

Figure 1: Causal DAG with observed confounders $O$.

generalized 1SDFIV (G-1SDFIV) regression, which is implemented in the same way as 1SDFIV regression except that $L_n(u, \theta)$ is re-defined as

$$L_n(u, \theta) = \beta_1 \text{MDD}_n(Y - u^\top \phi(X, \theta)|Z)^2 + \beta_2 R_n(u, \theta), \tag{10}$$

where $\beta_1 \geq 0$ and $\beta_2 \geq 0$ are two hyperparameter. When $\beta_2 = 0$, G-1SDFIV reduces to 1SDFIV. When $\beta_2 > 0$, G-1SDFIV can account for the loss of $R_n(u, \theta)$ for updating the estimator of $\theta$, so it could enhance the prediction capacity of model (3) while tolerating certain bias from the use of an endogenous variable $X$ in $R_n(u, \theta)$. In this paper, we set $\beta_1 = \beta_2$ to balance the two losses. Robustness checks on the choice of $\beta_1$ and $\beta_2$ are provided in Appendix A.

In many empirical tasks, observed confounders or covariates $O$ could appear in the causal model. See Figure 1 for an illustrating example of causal directed acyclic graph (DAG). Following Xu et al. (2021), we can use two different neural networks to learn treatment feature and covariate feature. To be specific, we set the structural function $g(X, O) = u^\top(\phi(X, \theta_X) \otimes \varphi(O, \theta_O))$, where $\phi(X, \theta_X)$ is the feature map of treatment $X$ indexed by $\theta_X$, $\varphi(O, \theta_O)$ is the feature map of covariates $O$ indexed by $\theta_O$, and $\otimes$ is the tensor product operator defined as $a \otimes b = \mathbf{vec}(ab^\top)$. Using a similar one-stage learning procedure for G-1SDFIV, we can estimate $u$, $\theta_X$, and $\theta_O$. See Algorithm 2 for details.

---

**Algorithm 2** 1SDFIV Regression with Covariates

---

**Require:** Data $(X_i, Y_i, O_i, Z_i)$; Regularization parameters $\lambda$; Initial estimators $\tilde{\theta}_{X*}$ and $\tilde{\theta}_{O*}$; Batch size $n_b$; Learning rate $\alpha$; Number of updates $T$; Add covariates to $R_n^{(n_b)}$ being $R_n^{(n_b)}(u, \theta_X, \theta_O)$; Add covariates to $L_n^{(n_b)}$ being $L_n^{(n_b)}(u, \theta_X, \theta_O)$.

**Ensure:** Estimated structural function $g(x, o)$

1: $(\tilde{\theta}_X, \tilde{\theta}_O) \leftarrow (\tilde{\theta}_{X*}, \tilde{\theta}_{O*})$
2: **repeat**
3:      Sample $n_b$ data $(X_i^{(b)}, Y_i^{(b)}, O_i^{(b)}, Z_i^{(b)})$.
4:      **for** $t = 1$ to $T$ **do**
5:          Return function $\tilde{u}^{(n_b)}$, i.e., the minimizer of $R_n^{(n_b)}(u, \tilde{\theta}_X, \tilde{\theta}_O)$ using $(X_i^{(b)}, Y_i^{(b)}, O_i^{(b)})$
6:          Update $\tilde{\theta}_O \leftarrow \tilde{\theta}_O - \alpha \nabla_{\theta_O} L_n^{(n_b)}(\tilde{u}^{(n_b)}, \tilde{\theta}_X, \tilde{\theta}_O)$
7:      **end for**
8:      **for** $t = 1$ to $T$ **do**
9:          Return function $\tilde{u}^{(n_b)}$, i.e., the minimizer of $R_n^{(n_b)}(u, \tilde{\theta}_X, \tilde{\theta}_O)$ using $(X_i^{(b)}, Y_i^{(b)}, O_i^{(b)})$
10:        Update $\tilde{\theta}_X \leftarrow \tilde{\theta}_X - \alpha \nabla_{\theta_X} L_n^{(n_b)}(\tilde{u}^{(n_b)}, \tilde{\theta}_X, \tilde{\theta}_O)$
11:      **end for**
12: **until** convergence
13: Compute $\tilde{u} := \tilde{u}^{(n)}$, the minimizer of $R_n(u, \tilde{\theta}_X, \tilde{\theta}_O)$ using entire dataset
14: **return** $\tilde{g}(x) = \tilde{u}^\top(\phi(x, \tilde{\theta}_X) \otimes \varphi(o, \tilde{\theta}_O))$

---

We should highlight that when $\mathbb{E}(\varepsilon|Z) = 0$, we have $\mathbb{E}(f(Z)\varepsilon) = 0$ for any function $f(Z)$. Thus, the idea of generalized method of moments (GMM) (Hansen, 1982) can also learn our causal model (3), based on a sequence of moments $E[f_j(Z)(Y - g(X)]$ for $j = 1, \dots, J$. Although statistically

efficient under optimal weighting, GMM requires an initial consistent estimator of model parameter to compute the weight matrix, creating a dependency on good initialization. Moreover, the choice of moment functions $f_j$ critically affects identification and performance, since insufficient moments can lead to weak identification. DeepGMM (Bennett et al., 2019) formulates IV estimation as a smooth zero-sum game between two neural networks: one estimating the structural function $g$, and the other selecting moment conditions. However, according to Xu et al. (2021), the performance of DeepGMM is even worse than that of DeepIV and DFIV in the benchmark datasets. See also our experimental results in Section 4.

### 3.2 ANALYSIS FOR CONVERGENCE

Assume $u_0$ and $\theta_0$ are the true values of $u$ and $\theta$ in causal model (3). To show the convergence of our estimated causal effect under 1SDFIV, we need some sufficient technical assumptions. The first assumption below requires (i) the consistency of $\tilde{u}$, and (ii) the uniform convergence of $L_n(\tilde{u}, \theta)$.

**Assumption 2** *(i) $\tilde{u} \xrightarrow{p} u_0^*$ as $n \to \infty$; (ii) $\sup_{\theta \in \Theta} |L_n(\tilde{u}, \theta) - L_n(u_0^*, \theta)| \xrightarrow{p} 0$ as $n \to \infty$.*

Denote
$$L_n^*(\theta) = L_n(u_0^*, \theta) = \mathrm{MDD}_n(Y - u_0^{*\top}\phi(X, \theta)|Z)^2$$
and $L^*(\theta) = \mathrm{MDD}(Y - u_0^{*\top}\phi(X, \theta)|Z)^2$. Following Song et al. (2025), we provide additional assumptions to ensure the consistency of $\tilde{\theta}$.

**Assumption 3** *The parameter $\theta_0^*$ is the minimal point globally identified by $\theta_0^* = argmin_{\theta \in \Theta} L^*(\theta)$.*

**Assumption 4** *Parameter space $\Theta$ is compact and $\theta_0^*$ is an interior point of $\Theta$.*

**Assumption 5** *$\{(X_i, Y_i, Z_i)\}$ is an i.i.d. sequence, and all entries of $Z_i$ are independent.*

**Assumption 6** *(i) $\phi(x, \theta)$ is twice continuously differentiable for each $(x, \theta) \in \mathbb{R}^{d_X} \times \Theta$; (ii) there exists a function $c(x)$ such that $\|\nabla_\theta \phi(x, \theta)\| \le c(x)$ and $\mathbb{E}\|c(X)\|^4 < \infty$; (iii) $\mathbb{E}\|Z\|^4 < \infty$.*

**Theorem 1** *Suppose Assumptions 1–6 hold. Then, $\tilde{\theta} \xrightarrow{p} \theta_0^*$ as $n \to \infty$. Moreover, we have*

$$\frac{1}{n}\sum_{i=1}^n \tilde{u}^\top \phi(X_i, \tilde{\theta}) \xrightarrow{p} \mathbb{E}[u_0^\top \phi(X, \theta_0)] \tag{11}$$

*as $n \to \infty$.*

From Theorem 1, we know that $\tilde{\theta}$ converges to a quasi-true value $\theta_0^*$, which is not necessarily the true value $\theta_0$. This is also the case for $\tilde{u}$, which is assumed to converge to a quasi-true value $u_0^*$ in Assumption 2. Although $(u_0^*, \theta_0^*)$ could be different from $(u_0, \theta_0)$, our averaged estimated causal effect in (11) converges to the true average causal effect $\mathbb{E}[u_0^\top \phi(X, \theta_0)]$.

It is worth noting that if $u$ and $\theta$ are jointly estimated using the MDD loss function $L_n(u, \theta)$ in (8), it is expected that their estimators can converge to the true values $u_0$ and $\theta_0$. However, this joint estimation method is neither computationally efficient nor well-suited to the mean squared error prediction criterion (without making use of $R_n(u, \theta)$).

## 4 EXPERIMENTS

In this section, we evaluate the empirical performance of our proposed methods, 1SDFIV and G-1SDFIV, through numerical experiments in both low- and high-dimensional settings, following the design in Xu et al. (2021). For the low-dimensional case, we use the *Air Ticket Demand* dataset, while for the high-dimensional case, we combine the Air Ticket Demand dataset with the *MNIST* dataset (LeCun & Cortes, 2010).

To highlight the advantages of our one-stage estimation framework, we compare 1SDFIV and G-1SDFIV with the following three baselines:

(1) **DFIV**: a two-stage method proposed by Xu et al. (2021), which predicts the potential outcome by using a neural network to construct feature map for IV in the first stage and another neural network to construct feature map for treatment in the second stage.

(2) **DeepIV**: a two-stage method proposed by Hartford et al. (2017), which predicts the potential outcome by fitting distribution of treatment given IV in the first stage and applying neural networks to describe the counterfactual prediction function in the second stage.

(3) **DeepGMM**: a GMM-based method proposed by Bennett et al. (2019), which predicts the potential outcome using the class of all neural networks of a given architecture with varying weights to create different moment restrictions.

Here, DFIV represents the current state-of-the-art and serves as the most closely related competitor to our methods. The key distinction lies in the estimation approach: our methods use an MDD-supervised one-stage procedure, while DFIV adopts a two-stage procedure. This comparison allows us to examine whether the one-stage approach leads to improved estimation performance.

To evaluate out-of-sample predictive power, we follow the existing literature (Hartford et al., 2017; Bennett et al., 2019; Xu et al., 2021) and use the mean squared error (MSE) on the test set as the performance metric. To account for randomness in the training process, each method is repeated with 20 different random seeds.

### 4.1 Low-dimensional Case

The Air Ticket Demand dataset consists of simulated samples $\{(Y, P, T, S, C)\}$, where $Y$ is number of ticket sales, $P$ is the ticket price, $T \in [0, 10]$ is the time of year, $S \in \{1, \dots, 7\}$ is the customer type (defined by different levels of price sensitivity), and $C$ the fuel price.

In this paper, we regard $Y$ as the outcome of interest, $P$ as the treatment, and $T$ and $S$ as two observed confounders. The noise terms in both $Y$ and $P$ are assumed to be correlated, indicating the influence of an unmeasured confounder, with the correlation strength determined by an endogeneity level coefficient $\rho \in [0, 1]$. To address the bias from this hidden confounder, the fuel price $C$ is used as an instrumental variable.

Specifically, we define the data-generating process as

$$Y = g(P, T, S) + \varepsilon \quad \text{with} \ \mathbb{E}[\varepsilon | C, T, S] = 0, \tag{12}$$

where $\varepsilon$ is the noise term, and $g$ is the structural function given by

$$g(P, T, S) = 100 + (10 + P)Sh(T) - 2P,$$

with

$$h(T) = 2\left[\frac{(T-5)^4}{600} + \exp(-4(T-5)^2) + \frac{T}{10} - 2\right].$$

Next, we generate simulated samples, $(P, T, S, C)$, by $S \sim \text{Unif}\{1, \dots, 7\}$, $T \sim \text{Unif}[0, 10]$, $C \sim \mathcal{N}(0, 1)$, $V \sim \mathcal{N}(0, 1)$, and $P = 25 + (C + 3)h(T) + V$. Meanwhile, we specify the noise term as $\varepsilon \sim \mathcal{N}(\rho V, 1 - \rho^2)$, making it dependent on both $V$ and the hyperparameter $\rho$. Given $(P, T, S, C)$ and $\varepsilon$, we compute the outcome $Y$ via (12), allowing us to simulate any number of samples for the training set.

Lastly, for the testing set, samples are generated with random noise terms but fixed choices of $P, T$, and $S$ (Xu et al., 2021). To be specific, we combine 20 evenly spaced values of $P \in [10, 25]$, 20 evenly spaced values of $T \in [0, 10]$, and all 7 values of $S \in \{1, \dots, 7\}$ to conduct these choices. This yields $20 \times 20 \times 7 = 2800$ distinct combinations of $(P, T, S)$, corresponding to 2800 test samples used to compute the out-of-sample MSE.

Figure 2 demonstrates the out-of-sample MSE across different combinations of endogeneity level coefficient $\rho \in \{0.1, 0.25, 0.5, 0.75, 0.9\}$ and (training) sample size $n \in \{1000, 5000, 10000\}$. From this figure, we can draw the following conclusions:

(1) 1SDFIV and G-1SDFIV consistently achieve lower MSE values than their competitors across different endogeneity levels and sample sizes, with DFIV ranking third. This finding

is not beyond our expected, since the one-stage approach directly exploits the exogeneity of the IV through the MDD loss, without first learning an exogenous variable from the IV, as required by the two-stage methods.

(2) When the sample size is small (e.g., 1000), G-1SDFIV outperforms 1SDFIV due to its superior capacity of dealing with mis-specification model. As the sample size becomes large (e.g., 5000 or 10000), the performance of 1SDFIV and G-1SDFIV tends to be similar.

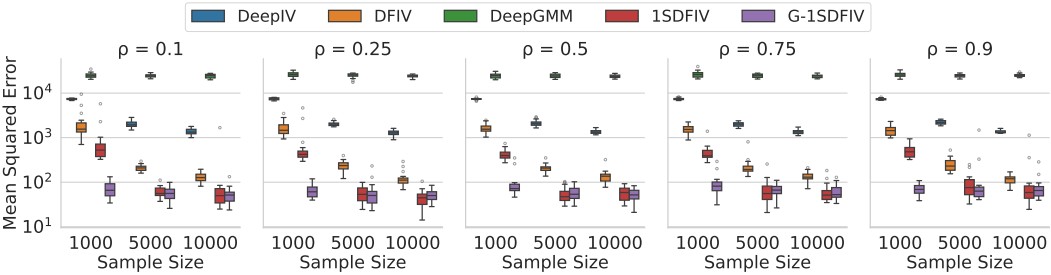

Figure 2: Out-of-sample MSE for Air Ticket Demand dataset with low-dimensional confounders.

## 4.2 HIGH-DIMENSIONAL CASE

In practice, discrete inputs such as the customer type $S$ in the previous subsection are often unavailable. Instead, the outcome of interest may depend on high-dimensional, unstructured inputs (e.g., images or behavioral traces) that implicitly encode such discrete information.

To evaluate our method in this setting, we replace the discrete input $S \in \{1, \ldots, 7\}$ in the Air Ticket Demand dataset with an image of the corresponding handwritten digit from the MNIST dataset (LeCun & Cortes, 2010)[1]. This replacement mimics scenarios where the true customer type is unobserved and must be inferred from noisy, high-dimensional data (Xu et al., 2021). Apart from this replacement, the rest of the data-generating process remains unchanged.

Figure 3 shows the out-of-sample MSE for all methods across different endogeneity levels, with the training sample size fixed at 5000. From this figure, we can see:

(1) G-1SDFIV consistently dominates all competitors, demonstrating clear advantages regardless of the choice of endogeneity level. Its significant performance is probably due to its superior capacity for characterizing/capturing/approximating causal relationship adequately.

(2) Although 1SDFIV presents strong theoretical properties, it performs slightly worse than DFIV and DeepIV. This finding may be attributed to the limited sample size, which is moderate to the dimensionality of confounders.

## 4.3 COMPUTATION BURDEN

To compare the computational burden, we report the total training time of each method in both low- and high-dimensional cases. As shown in Table 1, our proposed methods, 1SDFIV and G-1SDFIV, require significantly less training time than the existing approaches. This advantage arises directly from the efficiency of the one-stage estimation framework.

## 5 CONCLUSION

In this paper, we introduce two new methods 1SDFIV and G-1SDFIV to identify causal effects in a one-stage framework. The novelty of both methods is to use the MDD loss, which allows us to

---

[1]For example, if a simulated sample has $S = 1$, the original feature is replaced by a randomly selected image of the digit 1 from the MNIST dataset.

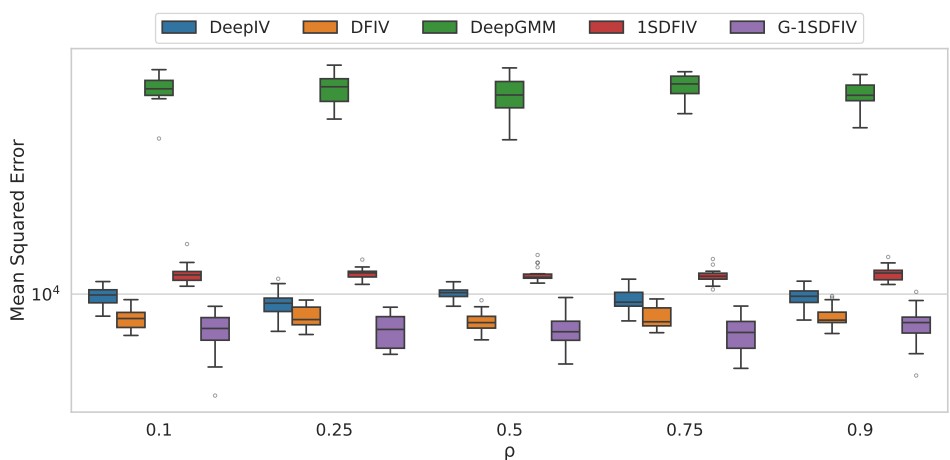

Figure 3: Out-of-sample MSE for Air Ticket Demand dataset with high-dimensional confounders.

Table 1: Average training time (in hours) across different methods.

| Method | Low-dimensional Case | High-dimensional Case |
|---|---|---|
| **1SDFIV & G-1SDFIV** | **2** | **7** |
| DeepIV | 4 | 14 |
| DFIV | 5 | 19 |
| DeepGMM | 9 | 27 |

directly estimate the structural function from the exogeneity of the IV. By circumventing the traditional two-stage implementation, 1SDFIV and G-1SDFIV mitigate error propagation, reduce computational burden, and lessen sensitivity to model mis-specification. Comprehensive experiments on both low- and high-dimensional datasets demonstrate that 1SDFIV and G-1SDFIV not only achieve higher prediction accuracy but also provide significant computational gains over existing two-stage methods. These results highlight the potential of direct MDD-based estimation as a more efficient and robust alternative to learn causal effects.

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

## A  HYPERPARAMETER TUNING

We mainly focus on tuning the ratio $\beta_1/\beta_2$ in formula (10). In both low-dimensional and high-dimensional settings, we take four different ratios at different combinations of sample size $n$ and and do 20 independent experiments under each ratio to evaluate the performance on validation set. Specifically, we train the model from the training dataset and evaluate the model performance via prediction mean squared error on the independent validation dataset. Then, we take the median result of the 20 independent experiments. For low-dimensional case, we validate several ratios under three sample sizes and five endogeneity levels. For high-dimensional case, we validate several ratios under endogeneity levels when sample size is $5000$.

Table 2: $\beta_1/\beta_2$ tuning in G-1SDFIV for low-dimensional data

| Configuration | Ratio | | | | |
|---|---|---|---|---|---|
| | 2:1 | 3:2 | 1:1 | 2:3 | 1:2 |
| $n = 1000$ | | | | | |
| $\rho = 0.10$ | 70.06 | 63.92 | 65.85 | 72.33 | 61.43 |
| $\rho = 0.25$ | 52.20 | 66.91 | 60.84 | 63.65 | 54.50 |
| $\rho = 0.50$ | 43.69 | 71.16 | 71.93 | 70.69 | 56.55 |
| $\rho = 0.75$ | 73.75 | 70.43 | 81.49 | 74.61 | 70.56 |
| $\rho = 0.90$ | 49.36 | 75.45 | 68.80 | 79.34 | 59.69 |
| $n = 5000$ | | | | | |
| $\rho = 0.10$ | 50.79 | 60.88 | 55.95 | 55.88 | 49.78 |
| $\rho = 0.25$ | 77.30 | 48.11 | 50.35 | 54.61 | 75.12 |
| $\rho = 0.50$ | 54.91 | 61.19 | 53.61 | 59.67 | 68.20 |
| $\rho = 0.75$ | 64.06 | 62.51 | 66.52 | 58.05 | 58.87 |
| $\rho = 0.90$ | 88.38 | 69.43 | 62.97 | 57.88 | 80.93 |
| $n = 10000$ | | | | | |
| $\rho = 0.10$ | 69.18 | 58.72 | 51.55 | 53.55 | 60.64 |
| $\rho = 0.25$ | 60.31 | 45.91 | 50.16 | 46.57 | 57.63 |
| $\rho = 0.50$ | 80.71 | 51.34 | 51.82 | 59.96 | 72.01 |
| $\rho = 0.75$ | 70.61 | 59.87 | 52.55 | 62.79 | 57.40 |
| $\rho = 0.90$ | 70.30 | 56.75 | 65.13 | 63.51 | 59.23 |

From Tables 2 and 3, we discover that the selection of ratio $\beta_1/\beta_2$ does not have a significant impact on the validation accuracy. So in this paper, we naturally chose this ratio to be $1$.

## B  TECHNICAL PROOF

**Proof of Theorem 1.** Recall $\tilde{\theta} = \arg\min_{\theta \in \Theta} L_n(\tilde{u}, \theta)$. By Assumption 2, we have

$$\tilde{\theta} = \tilde{\theta}^* + o_p(1),$$

Table 3: $\beta_1/\beta_2$ tuning in G-1SDFIV for high-dimensional data

| Ratio $\rho$ | 2:1 | 3:2 | 1:1 | 2:3 | 1:2 |
|---|---|---|---|---|---|
| 0.1 | 7358.86 | 7782.02 | 7501.98 | 7623.57 | 7135.79 |
| 0.25 | 7399.44 | 7245.2 | 7441.63 | 7541.95 | 7204.26 |
| 0.5 | 7074.47 | 7101.95 | 7299.58 | 7484.82 | 7124.68 |
| 0.75 | 7227.39 | 7153.21 | 7256.1 | 7454.14 | 7795.98 |
| 0.9 | 7211.23 | 7357.52 | 7881.85 | 7209.02 | 7354.05 |

where $\tilde{\theta}^* = \arg\min_{\theta \in \Theta} L_n(u_0^*, \theta)$. Next, by Assumptions 1 and 3–6, using a similar argument as for Theorem 2.1 in Song et al. (2025), we can show

$$\tilde{\theta}^* = \theta_0^* + o_p(1),$$

which entails $\tilde{\theta} \xrightarrow{p} \theta_0^*$ as $n \to \infty$.

Moreover, since $\tilde{u} \xrightarrow{p} u_0^*$ and $\tilde{\theta} \xrightarrow{p} \theta_0^*$ as $n \to \infty$, using the uniform law of large numbers theorem for the i.i.d. data, we have

$$\frac{1}{n} \sum_{i=1}^{n} \tilde{u}^\top \phi(X_i, \tilde{\theta}) \xrightarrow{p} \mathbb{E}[u_0^{*\top} \phi(X, \theta_0^*)] \tag{13}$$

as $n \to \infty$. On one hand, from the identification of $\theta_0^*$ in Assumption 3, we have $\text{MDD}(Y - u_0^{*\top}\phi(X, \theta_0^*)|Z)^2 = 0$, which indicates $\mathbb{E}(Y - u_0^{*\top}\phi(X, \theta_0^*)|Z) = 0$ by (7) and thus

$$\mathbb{E}(Y) = \mathbb{E}(u_0^{*\top}\phi(X, \theta_0^*)).$$

On the other hand, by Assumption 1, we have $\text{MDD}(\varepsilon|Z)^2 = \text{MDD}(Y - u_0^\top\phi(X, \theta_0)|Z)^2 = 0$. By (7), it follows that

$$\mathbb{E}(Y) = \mathbb{E}(u_0^\top\phi(X, \theta_0)).$$

Therefore, $\mathbb{E}(u_0^{*\top}\phi(X, \theta_0^*)) = \mathbb{E}(u_0^\top\phi(X, \theta_0))$. Together with (13), result (11) holds. This competes the proof. $\square$

## C  THE USE OF LARGE LANGUAGE MODELS (LLMS)

During the writing of this manuscript, we employed Large Language Models (LLMs) to assist with refining the text. Their main role was to enhance the manuscript's readability and clarity by fixing grammatical mistakes and refining sentences. All substantive scientific aspects—such as the development of methods, experimental planning, and interpretation of findings—were solely conducted by the authors. The LLMs were used purely as tools for improving the writing and did not influence the research concepts or results described in the paper.

