# OpenReview forum: "One-stage Deep Feature Instrumental Variable Regression"
_ICLR.cc/2026/Conference — Submitted to ICLR 2026_

### Official Review · Reviewer_DYio · 2025-10-15

**Soundness:** 2
**Presentation:** 2
**Contribution:** 2
**Rating:** 2
**Confidence:** 5

**Summary:**

This manuscript proposed a one-stage method for instrument variable regression based on the usage of martingale difference divergence.

**Strengths:**

Introduce martingale difference divergence into this problem.

**Weaknesses:**

## Methodology
The authors missed one key references [1], which also proposed a one-stage estimation method for instrument variable regression, which can be viewed as leveraging certain function class to perform conditional independence test that shares the same spirit to this manuscript. There are also lots of following work on this direction (e.g. [2] to accommodate observable confounders). In my opinion, the authors only use the martingale difference divergence to perform conditional independence test, which limits the novelty of this paper.

[1] Muandet, Krikamol, et al. "Dual instrumental variable regression." Advances in Neural Information Processing Systems 33 (2020): 2710-2721.
[2] Sun, Haotian, et al.  "Spectral Representation for Causal Estimation with Hidden Confounders." International Conference on Artificial Intelligence and Statistics. PMLR, 2025.

## Theoretical Analysis
Only asymptotic convergence results are shown in the paper. Meanwhile, paper like [1] has non-asymptotic error rate based on standard assumptions, which makes me feel the theoretical analysis in this paper is not so meaningful.

## Experimental Results
The authors only conduct experiments with relatively toy dataset. For high-dimensional confounders it does not significantly outperforms DFIV, which makes me concerning about applicability for high-dimensional scenarios.

**Questions:**

To echo the weakness part:

* How to compare the proposed method with dual-IV based method?
* Are there any theoretical benefits compared with dual-IV based method?
* How's the empirical performance compared with dual-IV based method?

---

### Official Review · Reviewer_VN85 · 2025-10-29

**Soundness:** 3
**Presentation:** 2
**Contribution:** 2
**Rating:** 2
**Confidence:** 4

**Summary:**

This work introduces a single-stage instrumental variable regression framework that extends DFIV by employing a loss function based on the Martingale Difference Divergence (MDD) measure. The proposed approach effectively unifies the two-stage estimation process and achieves superior performance compared to the baseline methods in the experiments.

**Strengths:**

The proposed approach effectively unifies the two-stage estimation process and achieves superior performance compared to the baseline methods in the experiments.

**Weaknesses:**

1.	The description of the main challenges in the introduction is somewhat inaccurate (line 062-069), which weakens the motivation of the paper.
2.	The manuscript contains some subjective expressions and minor inaccuracies in technical terminology. It would be beneficial for the authors to refer to the single-stage estimation process introduced in prior work on proximal inference [1], which could help enhance the precision and objectivity of the presentation.
3.	The novelty of the proposed method appears to be limited.

**Questions:**

Major questions:

>The literature review is not sufficiently comprehensive. Prior studies [2] have proposed improvements to the two-stage structure of DeepIV, developing a single-stage Deep IV. This line of work should be cited and included as a baseline in the experiments. In addition, due to the existence of such work, certain statements in the manuscript are not entirely accurate—for example, the sentence on line 142: “Note that these two issues also exist in DeepIV”, the sentence on line 051: “DeepIV suffers from two practical drawbacks.”

>Regarding the motivating challenges (line 062-069), the manuscript identifies two main issues: (1) both DeepIV and DFIV require carefully tuned hyperparameters, and (2) they do not generalize well. However, in the proposed method, I noticed that (1) the hyperparameter settings appear to be almost identical to those used in DFIV, and (2) the generalization performance does not seem to clearly surpass that of DFIV. These points should be further clarified and justified in the revision.

Minor questions:

>The manuscript repeatedly refers to X ̂ as a “learned exogenous variable.” However, X is an endogenous variable in the IV framework [3, 4], and X ̂ represents its predicted (or fitted) value obtained from the first stage.

>The References section should start on a new page.



[1] Kompa B, Bellamy D, Kolokotrones T, Beam A. Deep learning methods for proximal inference via maximum moment restriction. Advances in Neural Information Processing Systems. 2022 Dec 6;35:11189-201.

[2] Lin A, Lu J, Xuan J, Zhu F, Zhang G. One-stage deep instrumental variable method for causal inference from observational data. In2019 IEEE International Conference on Data Mining (ICDM) 2019 Nov 8 (pp. 419-428). IEEE.

[3] Bun MJ, Harrison TD. OLS and IV estimation of regression models including endogenous interaction terms. Econometric Reviews. 2019 Aug 9;38(7):814-27.

[4] Bound J, Jaeger DA, Baker RM. Problems with instrumental variables estimation when the correlation between the instruments and the endogenous explanatory variable is weak. Journal of the American statistical association. 1995 Jun 1;90(430):443-50.

---

### Official Review · Reviewer_Leen · 2025-10-30

**Soundness:** 2
**Presentation:** 2
**Contribution:** 3
**Rating:** 4
**Confidence:** 5

**Summary:**

Instrumental variable approach is a fundamental tool for mitigating the influence of unobserved confounders. Traditional IV methods commonly follow a two-stage learning strategy, but such two-stage approaches suffer from inherent problems including error propagation, high computational cost, and sensitivity to model misspecification. To address these issues, the authors propose a novel one-stage deep feature IV method, 1SDFIV. 1SDFIV constructs its loss function based on the martingale difference divergence (MDD) and directly learns the causal structural function. Furthermore, the authors introduce a generalized regression variant, G-1SDFIV, to improve predictive performance and robustness.

**Strengths:**

1.The method is simple and constructs the loss function directly using an MMD-based measure.

2.The authors design the generalized version G-1SDFIV to mitigate potential drops in predictive performance for 1SDFIV in small-sample or model-misspecification scenarios.

3.The approach has low computational overhead.

**Weaknesses:**

1.Readability needs improvement. I recommend the authors add illustrative figures in appropriate places to show (a) the workflow of two-stage learning strategies, (b) the workflow of 1SDFIV, and (c) the differences between 1SDFIV and existing two-stage approaches.

2.The role of Equation (8) is unclear. The authors claim that Equation (8) is evidently a single-stage formulation. However, from Equation (8) it appears the authors may have simply skipped the step of fitting the exogenous variables. Equation (8) looks more like: given Z, compute a loss between the model’s prediction and the observed outcome Y, where that loss happens to be an MMD. This point requires clarification.

3.On the use of MMD to capture dependence. In Section 2.2 the authors claim that MMD can capture mean dependence between random variables. Is it then appropriate to use MMD to measure dependence between the instrument Z and the treatment X? By definition of an instrument, X and Z are correlated, so the feasibility and implications of this choice should be discussed.

4.Comparison to SOTA baselines is missing. The authors should compare their method against state-of-the-art baselines to demonstrate superiority. Currently the newest baseline used is DFIV from 2021.

5.Typesetting errors in equations. Some equations are missing equation numbers and should be corrected.

Note that: the convergence theorem (Theorem 1) only ensures that the estimated parameters converge to “quasi-true” values but does not establish asymptotic unbiasedness or consistency for the causal function g(x).

**Questions:**

How does minimizing MDD ensure that E[Y∣Z]=E[g(X)∣Z] holds when g(X) is nonlinear?

Can the method fail under weak instruments or near-collinearity?

What is the computational complexity of computing the empirical MDD?

Since 1SDFIV uses deep neural feature mappings, how can one interpret the learned causal mechanism?

---

### Official Review · Reviewer_gZXS · 2025-10-30

**Soundness:** 1
**Presentation:** 2
**Contribution:** 1
**Rating:** 2
**Confidence:** 3

**Summary:**

The authors propose a nonparametric instrumental variable estimator based on deep neural network features of the instrument and treatment variables.  The estimator is based on minimising a linear combination of a Martingale Difference Divergence and a squared-error regression loss. The key advantage of the method, as presented by the authors, is that it has a lower computational overhead than deep-learning-based alternatives, such as DeepIV (Hartford et al. 2017) and DFIV (Xu et al. 2021). Experiments and a rudimentary theory back the proposed method.

**Strengths:**

The authors correctly identify that both DeepIV and DFIV can be computationally demanding, and their practical performance hinges on careful hyperparameter tuning and possibly long training.

The proposed methods and algorithms are described clearly.

The preliminary section and background information on DFIV are largely well-written.

**Weaknesses:**

1. If Equation 7 is correct, then $\mathrm{MDD}(Y-u^T\phi(X,\theta),Z)^2=0$ does not imply that $u^T\phi(X,\theta)$ is the correct structural function. For instance, one could add any constant to it and still have $\mathrm{MDD}(Y-u^T\phi(X,\theta)+c, Z)^2=0$. Hence, just minimising Equation 8 does not guarantee the identification of the correct parameters. Instead of doing so, the authors propose to alternate between gradient descent steps on Equation 8 (the MDD) and explicit optimisation of Equation 9 (the regression loss). This decision is motivated by the direct optimisation of (8) being “time-consuming”. However, in light of the observation above, the more important reason seems to be that if $f$ is the true structural function and $u^T\phi(X,\theta)=f+c$ for some constant $c$, then the regression loss is in fact minimised by having $c=0$ (but it also encourages the modification of $f$ towards $\mathbb E[Y|X]$). The minimisation of (9) is not primarily dictated by computational considerations but by the fact that minimising (8) alone has no reason to recover the correct function. Regardless, including the minimisation of (9) in the procedure raises doubts about the population-level optimum of the procedure and whether it has anything to do with the IV structural function. The same issue is even clearer if one optimises (10) with non-decaying $\beta_2$ (in this work, $\beta_1=\beta_2$ is constant). A minimum of the regression loss of $Y$ on $X$ is not the IV structural function, so it is not clear why the solution to (10) should yield anything close to an IV solution.

2. The proof of Theorem 1 is wrong or written in an unclear way. The conclusion in lines 562-564 seems incorrect. $\mathrm{MDD} (Y-u_0^{T} \phi(X,\theta_0)|Z)=0$ does not imply $\mathbb E[Y-u_0^{T}\phi(X,\theta_0)|Z]=0$. It only implies that the expectation is constant. Assuming this is not the case and it is Equation 7 that is wrong, Assumption 2 looks a priori too strong to consider the theoretical result substantial either way.

3. The validation procedure for the $\beta_1/\beta_2$ ratio seems to depend on the unobserved structural function values. If that is the case, it is impossible to execute in practice.

4. The characterisation of DFIV in Equation 4 is incorrect. There should be a feature map applied to $X$.

5. The “correct” approach to DFIV was proposed by Petrulionyte et al. 2024 (Functional Bilevel Optimization for Machine Learning). A comparison to DFIV, as performed by the authors of that work, would be preferable to the current benchmark.

6. A discussion of important recent references on nonparametric IV is missing. E.g. Dikkala et al. 2020 (Minimax Estimation of Conditional Moment Models) and Li et al. 2024 (Regularized DeepIV with Model Selection).

**Questions:**

1. Can the authors comment on why/whether the population-level minimiser of Equation 8 should be a solution to the IV inverse problem? Furthermore, why does including the regression loss with a non-decaying weight in (10) not affect identifiability?

2. How is minimising the MDD loss with gradient descent less efficient than alternating between MDD gradient updates of $\theta$ and closed-form updates for $u$ based on the regression loss?

3. Can the authors please elaborate on the meaning of lines 213-215? I do not understand what “causal effect has a correct specification different from that of model (3)” means.

4. What is the meaning of the following in lines 311-314? “However, this joint
estimation method is neither computationally efficient nor well-suited to the mean squared error
prediction criterion”. If the parameters obtained by the minimisation of $L_n$ can converge to the true data-generating ones, then surely $L_n$ is a good criterion to minimise?

5. Is my understanding of the $\beta_1/\beta_2$ validation procedure, as described above, correct?

6. Why is the reasoning in lines 562-564 correct?

7. What is an example of a setting in which the assumptions of Section 3.2 are satisfied?

---

### Meta-Review · Area_Chair_S1HQ · 2026-01-06

**Summary:**

Various concerns raised in the initial reviews
- Mathematical errors
- Lack of discussion on recent related works
- Poor readability
- Missing comparison to SOTA baselines
- Limited novelty
- Not sufficient theoretical analysis
- Limited empirical evaluation

**Reviewer Concerns:**

The authors did not provide a rebuttal.

**Reviewer Scores:**

Scores (2, 4, 2, 2) will not change due to lack of rebuttal.

---

### Decision · Program_Chairs · 2026-01-26

Reject